# Under nutrition and associated factors among adolescent girls attending school in the rural and urban districts of Debark, Northwest Ethiopia: A community-based comparative cross-sectional study

**Tewodros Getaneh Alemu** *, **Addis Bilal Muhye, Amare Demsie Ayele**

Department of Pediatrics and Child Health Nursing, School of Nursing, College of Medicine and Health Sciences and Specialized Hospital, University of Gondar, Gondar, Northwest Ethiopia

* tewodrosgetaneh7@gmail.com

**Data Availability Statement:** All relevant data are within the manuscript and its Supporting Information files.

## Abstract

### Introduction

Adolescence is the time of puberty in which a substantial changes in physical, mental, and emotional are observed; Nutritional requirements significantly rise as a result. Even though improving adolescent girls' nutritional status helps to break the intergenerational cycle of malnutrition, many studies in Ethiopia focus on determining the nutritional status of under-five and pregnant women and even they don't show disparities between urban and rural adolescent girls. Thus, this study was aimed at comparing the rural and urban prevalence's of stunting and thinness and their associated factors among adolescent girls attending school in Debark district, Northwest Ethiopia, 2020.

### Method

A community-based comparative cross-sectional study was conducted among 792 adolescent girls from February 25th to March 21st 2020. A multi-stage sampling followed by simple random sampling technique was used. Data were collected through a face-to-face interviewer-based questionnaire. Different anthropometric measurements were taken. The collected data were entered to Epi-data and exported into SPSS for analyses. Variables with p-values < 0.25 in the bivariable analysis were exported to multivariable logistic regression model to control confounders and identify the factor. The strength of association and statistical significance was declared using the adjusted odds ratios with its corresponding 95% CI, and p-value $\leq 0.05$ respectively.

### Results

A total of 757 adolescent girls with a response rate of 95.6% were participated in the study. The overall prevalence of stunting and thinness were 20.1% and 10.3%, respectively. Stunting among rural adolescent girls was 24.2%; whereas it was 16% among urban residents.

**Funding:** the author(s) received no specific fund for this work.

**Competing interests:** No authors have competing interests.

**Abbreviations:** BMI, Body Mass Index; CM, Centimeter; DDS, Dietary Diversity Score; EDHS, Ethiopian Demographic and Health Survey; FAO, Food and Agricultural Organization; FMOH, Federal Ministry of Health; HFIAS, Household Food Insecurity Access Scale; KG, Kilogram; KMO, Keiser Meyer Olkin; MDD, Minimum Dietary Diversity; NNP, National Nutrition Program; PCA, Principal Component Analysis; SDG, Sustainable Development Goal; WHO, World Health Organization.

Likewise, thinness among rural adolescent girls was 8.5%; whereas it was 12.1% among urban dwellers. No latrine [AOR: 1.95 (95% CI: 1.11, 3.43)], lowest media exposure [AOR: 5.14 (95% CI: 1.16, 22.74)], lower wealth class [AOR:2.58 (95% CI: 1.310, 5.091)], and middle wealth class[AOR: 2.37 (95% CI: 1.230, 4.554)] have risen the likelihood of stunting in rural setting while early adolescent age [AOR:3.17 (95% CI:1.445,6.95)] significantly associated with stunting in urban setting. Food insecurity [AOR: 1.95 (95% CI: 1.01, 3.78)] was associated with stunting in overall adolescent girls. Middle adolescent age groups in rural area have more than three times to experience thinness [AOR: 3.67 (95% CI: 1. 21, 11.149)]. Whereas urban resident girls fall in early adolescent age group developed thinness were eight times [AOR: 8.39 (95% CI: 2.48–28.30)].

## Conclusion

Stunting was higher among rural adolescent girls as compared to urban. However, thinness was higher among urban dwellers. Lower wealth class, food insecurity, lowest media exposure, and age were significantly associated with stunting and thinness. Hence, increasing latrine coverage, boosting the economic status of the community, and increasing media exposure for adolescent girls should get due attention.

## Introduction

The World Health Organization (WHO) defines, adolescent as the age between 10–19 years [1]. At the end of 2018, there were 1.2 billion adolescents that accounts one-fourth of the sub-Saharan Africa population [2].

In Ethiopia, children and adolescents together constitute about 48% of the population, of which one-quarter is adolescent girls [3]. Every year, more than 1.2 million adolescents die worldwide and nutrition related problem is the foremost risk factor contributing to many of the main causes of death [2]. Adolescence is the time of puberty in which there are dramatic physical, mental, and emotional alterations fundamentally observed which demand various nutrients [4].

During adolescence, the growth rate is faster than growth rate seen at any other time in the person's life next to the infantile period [5]. Likewise, the hormones mediating the pubertal growth, pubertal sex hormones, and growth hormones normally rise together and are responsible for the optimal skeletal growth and sexual development. Which are modulated to a great extent by nutritional factors [5–7].

Globally, although there is a downward trend of stunting, still 130 million children and adolescents are suffering [8]. In low and middle-income countries, 67% of rural and 50% urban adolescents had a higher magnitude of underweight/thinness [9]. According to the Ethiopia Demographic and Health Survey (EDHS) 2016 report, the prevalence of thinness among Ethiopian adolescent girls was 29% [10]. Indeed, some other local studies have also revealed that the prevalence of stunting ranges from 12.5%-47.4% [11, 12], whereas thinnest is between 12.6% and 58.3% [13, 14].

Under-nutrition at this stage of life would continue to the next generation and shares the consequences to their children including low birth weight, short stature, and low resistance to infection [15–18]. This problem would be devastating in Ethiopia, where about 27.7% adolescent girls become pregnant at the age of 19 [19].

Adolescents who participated in sports, have a chronic illness, in need of excessive diet, or those who use alcohol and drug have a special nutrient requirement. For these reasons, this age group is nutritionally vulnerable [20–22]. Moreover, age, family size, maternal educational status, family income, frequency of feeding, and low dietary diversity score were frequently stated factors that affect the nutritional status of the adolescent girls [23–25]. In cognizance of this problem, the government of Ethiopia has initiated Growth and Transformation Plan II, Second National Nutrition Program (NNP II), and Seqota Declaration to fight against nutritional problems of the country. Nonetheless, the prevalence of under-nutrition is still high.

Even though improving adolescent girl's nutritional status helps to break the intergenerational cycle of malnutrition. Many studies in Ethiopia focus on under five and pregnant women nutrition and consider adolescents are low risk for under-nutrition. Previous evidences didn't show the clear disparities of under-nutrition between urban and rural adolescent girls. Moreover, nearly 80% of Ethiopian population have been living in the countryside areas in which there is high fertility rates and low socioeconomic status. Therefore, this study was aimed at comparing stunting and thinness in rural and urban adolescent girls and their associated factors at the district of Debark, Northwest Ethiopia.

## Methods and materials

### Study design, period and area

A community-based comparative cross-sectional study was conducted among rural and urban adolescent girls in the district of Debark from February 25th, to March 21st, 2020. Debark district is found in North Gondar Zone of Amhara National Regional State. It is located 830 km far from Addis Ababa, the capital city of Ethiopia. It has a total of 30 kebeles. Based on the 2007 census report, the woreda has a total population of 159,193. of whom 80,274 are men and 78,919 women; 138, 354 (87%), and 20,839 (13%) are rural and urban inhabitants, respectively [3]. A total of 11807-second cycle primary and secondary school adolescent girl students were attending school.

### Population

All school-going adolescent girls from a rural and urban school of Debark districts registered during the academic year 2019/2020 were the study participants. Adolescent girls who were from randomly selected schools in both areas in Debark district and available during the data collection period were included. However, adolescent girls with spinal curvature and other physical disabilities were excluded as there is no means to measure the anthropometrics.

### Sample size determination, technique, and sampling procedure

The sample size for each dependent variables (stunting and thinness) was calculated using the formula for a double population proportion by taking the required statistical assumptions: $Z\alpha/2 = 1.96$, power $(\beta) = 0.84$, r = ratio of $N_2$ to $N_1$ which is taken as one to one, and the sample size was 792 and 664 respectively. Therefore, is the larger sample size was 792. The calculated sample size was distributed equally among the two populations: i.e. 396 for rural and 396 for urban schools.

A multistage sampling technique was used. Initially, all primary second cycle and secondary schools (51 schools) in Debark district were stratified to urban and rural schools. Once again, stratification was done to each area by the school level (primary and secondary). A simple random sampling technique was used to select 20% of the 51 schools for each; accordingly, 9 schools from 44 rural schools and 4 schools from 7 urban schools were selected. From each of

the 13 schools, all sections of grades from 5–12 were separately enlisted and again 20% of the section from each grade was selected by simple random sampling. A total of 4979 adolescent girls were enumerated from the recruited schools. The student's classroom register was used as a sampling frame. About 396 from rural and 396 from urban resident girls were selected. Proportional allocation was done to select the number of students who participated from the selected grade. Finally, to select 792 study participants, a simple random sampling technique was used from the selected sections.

## Study variables

Stunting and thinness were the outcome variables and the socio-demographic variables were: age, the grade of the study participants, religion, family size, income, educational status of a mother, educational status of a father, occupation of a mother, occupation of a father, residence; dietary related factors: meal frequency, dietary diversity, food security; health and nutrition information factors: mass media exposure; health-related: menstruation and diarrhea; and household environment: the source of drinking water, availability of waste disposal, availability of latrine facilities, water treatment, water source distance, hand washing were independent variables.

## Definition of variables

➤ **Thinness:** BMI-for-age $<-2Z$ scores [26].

➤ **Stunting:** height-for-age $<-2Z$ scores [26].

➤ **Adolescents:** individuals in the age group of 10–19 years of age [27].

  • **Early adolescents:** adolescents in the age group of 10–13 years of age.

  • **Middle adolescents:** adolescents in the age group of 14–16 years of age.

  • **Late adolescents:** adolescents in the age group of 17–19 years of age.

➤ **Poor dietary diversity:** adolescent girls who consumed ($<5$ food groups) from the ten food items in the previous 24-hours [28].

➤ **good dietary diversity:** adolescent girls who consumed ($\geq 5$ food groups) from the ten food items in the previous 24-hours [28].

➤ **Food security:** Based on the Household Food Insecurity Access Scale (HFIAS) score (0–27) of 0–1 is categorized as food secure [29].

➤ **Food insecurity:** Based on (HFIAS) score (0–27) of 2 and above will be considered as food insecure [29].

➤ **Improved source of water**: Include piped water, public taps, standpipes, tube wells, boreholes, protected dug wells and springs, and rainwater [10].

➤ **Non-improved source of water:** including unprotected spring and unprotected well [10].

➤ **The variable mass media exposure** was constructed from the frequencies of reading newspapers/magazines, watching television, and listening to radio together [30]. Accordingly:

  ✓ **Not exposed**: All the three indicators are zero (Not at all)

✓ **Lowest exposed**: the highest value of all the three indicators is 1, i.e., exposed less than once a week to any one or all of the indicators.

✓ **Moderately Exposed**: the highest value of all the three indicators is 2, i.e., exposed at least once a week to any one or all of the indicators.

✓ **Very often exposed:** the highest value of all the three indicators is 3, i.e., very often exposed to any one or all of the indicators.

➢ **Urban:** per the Ethiopian urban planning law, an urban center is a locality with a minimum population size of 2000 inhabitants, at least 50% of whom are engaged in non-agricultural activities [31].

➢ **Rural:** formed by the inhabitants of a given area whose members are engaged either in agricultural and/or non-agricultural activities [31].

➢ **Water treatment:** adding bleach, boiling, filtration, if the household used either of them were taken as using appropriate water treatment methods, and if the household used neither of them considers inappropriate water treatment [32].

## Data collection tool and procedure

A structured face-to-face interviewer-administered questionnaire along with anthropometric measurements was used. The tool comprised of information on socio-demographic, household environmental factors, health, and nutrition information factors, dietary-related factors, household income. Stadiometers with a sliding headpiece attached to a digital weight scale were used to measure height and weight, respectively. Height was measured to the nearest 0.1cm while weight to the nearest 0.1kg in standing position. Each participant was weighed with minimum clothing and no footwear. To avoid variability among the data collectors, the same specifically trained measurers were employed for anthropometric measurement.

The study participant's wealth status was assessed by using questions adopting from Ethiopian Demographic and Health Survey, 2016 (EDHS). After labeling the variables between '0' and '1', the Principal Component Analysis (PCA) was applied. Then, the wealth status was ranked from the highest to the lowest.

Minimum dietary diversity (MDD) was assessed by interviewing participants to report how many of the 10 food items they consumed in the previous 24 hours before the survey. Dietary diversity questionnaires were prepared based on Food and Agricultural Organization (FAO) guideline 2016 [28]. Participants were informed to report as 'Yes' if she has eaten at least one food items, and coded as '1'. On the contrary, if the interviewee reports never consumed any food items, it was coded as'0'. Then, all responses were pooled and the variable, MDD was generated.

Household food security was classified based on responses to the nine severity items in the HFIAS. The procedure for scoring was used as follows: "0" was given if the event described by the question never occurred, "1" if it occurred during the previous 30 days. With regard to the occurrence, "1" was allotted if the events rarely occur, "2" sometimes and "3" often. Therefore, responses on the nine HFIAS questions were summed up to create a household food security score, with a minimum of "0" and a maximum score of "27."

Furthermore, two supervisors who have second-degree in pediatric nursing and six data collectors who have first-degree in nursing were involved in the data collection process. The data were collected from the schools where students are found and in their residential household in the community.

## Data quality control

The questionnaire was prepared first in English and then translated to the local language—Amharic and back to the English language to check its consistency. It was reviewed by language experts for translation of the language, and face validity was done by three public health nutritionists to check it's appropriateness for assessing stunting, thinness, and the independent variables. Pre-test was done among the 5% of the samples in rural and urban schools around Gondar Zuria school and community before the actual data collection was commenced and little amendments were done based on the pre-test result. The scales were carefully handled and periodically calibrated by placing standard calibration weights of 2kg iron bars on the scale. Furthermore, social desirability bias was attempted to reduce through interviewing the participants relatively in privacy secured area. Likewise, recall bias was tried to minimize by asking what kind of food items they consumed in each mealtime (breakfast, lunch, snack, and dinner). The collected data were checked out for its completeness during data collection by the principal investigator and supervisors.

## Data processing and analysis

Data were cleaned, coded, and entered to EPI data version 4.6 and exported to SPSS version 20 statistical packages for statistical analyses. Descriptive statistics were used to describe the socio-demographic characteristics of the respondents. Anthropometric measurements were converted to composite variables: height-for-age and BMI-for-age WHO $Z$ scores by using Antro Plus software. Flags were corrected to be between +6 and -6 after sorting the scores of individuals. A score of $<$-2 SD of BMI-for-age and height-for-age was coded as '1' and leveled as thin and stunted, respectively. On the other hand, a score of $>$-2 SD of BMI-for-age and height-for-age was coded as '0' and leveled as not thin and stunted, respectively.

The income status was analyzed through PCA. All categorical and continuous variables were categorized to be between '0' and '1'. Statistical assumptions of factor analysis were checked by Keiser-Meyer-Olkin (KMO) and Bartlett's test of sphericity. Accordingly, KMO of 0.7 or greater and significant Bartlett's test of sphericity (p-value<0.05) was considered as satisfying the assumption. In addition, communality value and Eigenvalues of 0.5 and greater and 1 respectively were included in factor analysis. Any variable with less than 0.5 of the communality value was removed from the analysis and the analysis was done repeatedly until all variables meeting the inclusion criteria for factor analysis. Next, all eligible factor scores were computed using the regression-based method to generate one variable, wealth status. Then after, the loading factors were sorted in their descending order and they were corrected to be between four and negative four. Following this, the final scores were ranked into three quantiles namely: first, second, and third. Finally, ranks were interpreted as upper class, middle class, and lower class respectively.

Furthermore, the binary logistic regression model was used to assess the association between dependent and independent variables for each outcome. A variable with a p-value of less than 0.25 was considered for multivariable logistic regression analysis. In the multivariable analysis, a p-value of less than 0.05 and an odds ratio with 95% CI were used to declare the presence and the strength of association between the independent and outcome variable. The Hosmer and Lemeshow test were used to diagnose the model fitness and the model was adequate. Finally, the result of this study was described in texts and tables.

## Ethical considerations and conscent for participants

Ethical clearance was obtained from College of Medicine and Health Sciences, School of Nursing on the behalf of the Institutional Review Board (IRB) of University of Gondar. Then, a

permission letter written from the Debark education office to the selected schools. Finally an official permission letter was obtained from each schools to proceed with the data collection. Written consent was obtained from the study participants by sending a letter to parents/guardians for each study participants less than 18 years. Then, assent was obtained from participants, and written consent was obtained from those ages greater than and equal to 18 years. Participation was after both informed written consent and assent were obtained. Voluntary engagement and confidentiality were retained to facilitate detailed and truthful self-disclosure.

## Results

### Socio-demographic characteristics

A total of 757 adolescent girls with a response rate of 95.6% (50.3% from urban and 49.7% from rural) were participated. The mean age of the adolescent girls was 14.7 (± 2.29SD). Regarding family size, 42.1% of the participants were in urban and 33.1% in rural schools live in a family size of five and below. About 10.9% of urban and 27.7% of rural participants were from lower wealth class (**Table 1**).

### Nutritional, health-related and environmental characteristics

The majority (84%) of girls from urban and a quarter (25%) in rural were from food secured. Less than three-quarters (70.9%) of them from urban and close to a quarter (24.7%) from rural had received adequate dietary diversity. (63.5%) of the adolescent girls in urban and (57.2%) of the in rural had eaten three and above times in a day. The majority (88.5%) of the adolescent girls from urban and, near to half (45.7%) of them from rural had a home latrine (Table 2).

### Prevalence of undernutrition

The overall magnitude of stunting among adolescent girls in schools of Debark district was 20.1% (95% CI: 17.3–23). The magnitude of stunting (24.2%) was higher among rural than urban school adolescent girls (16%). And the overall prevalence of thinness among adolescent girls in schools of Debark district was 10.3% (95% CI: 8.2%-12.7%). The prevalence of thinness was higher among urban adolescent girls (12.1%) than rural (8.5%).

### Factors associated with stunting

The regression model of rural adolescent girls suggested that the odds of being stunted among adolescent girls who had no latrine at home is increased by 95% compared to those who had home latrine [AOR: 1.95 (95% CI: 1.11, 3.43)]. Adolescent girls who had the lowest media exposure are 5.1 times higher to develop stunting compared to those who have very often media exposure [AOR: 5.14 (95% CI: 1.16, 22.74)]. Adolescent girls who were in the lower and middle wealth class group were 2.5 [AOR: 2.58 (95% CI: 1.31, 5.09)] and 2.3 [AOR: 2.37 (95% CI: 1.23, 4.55)] times higher to experience stunting, respectively as compared to adolescent girls in the upper wealth class group (**Table 3**).

Urban adolescent girls who were in the early adolescent age were 3 times higher to experience stunting as compared to adolescent girls in the late adolescent age [AOR: 3.17 (95% CI: 1.45, 6.95)] (**Table 4**).

The regression model of overall adolescent girls revealed that the odds of stunting among adolescent girls from food insecure household was increased by 95% compared to those from food secured households [AOR: 1.95 (95% CI: 1.01, 3.78)]. Those adolescent girls who were at the middle-age stage were 3 times more likely to be stunted as compared to late adolescent stage at first menstruation [AOR: 3.05 (95% CI: 1.22, 7)]. Adolescent girls who were in the

Table 1. Socio-demographic characteristics among school adolescent girls in Debark district, 2020 (n = 757).

| Variables | Response | Urban (381) Rural (376) | | | |
|---|---|---|---|---|---|
| | | Frequency | % | Frequency | % |
| Age | Early adolescent | 142 | 37.3 | 106 | 28.2 |
| | Middle adolescent | 146 | 38.4 | 164 | 43.6 |
| | Late adolescent | 93 | 24.4 | 106 | 28.2 |
| Level of education | Primary | 223 | 58.5 | 293 | 77.9 |
| | Secondary | 158 | 41.5 | 81 | 21.5 |
| Religion | Orthodox | 320 | 84 | 374 | 99.5 |
| | Muslim | 61 | 16 | 2 | 0.5 |
| Marital status | Single | 375 | 98.4 | 352 | 93.6 |
| | Married | 6 | 1.6 | 24 | 6.4 |
| Father's educational status | Unable to read and write | 23 | 6 | 128 | 34 |
| | Read and write | 111 | 29.1 | 191 | 50.8 |
| | Primary school | 91 | 23.9 | 44 | 11.7 |
| | secondary school | 64 | 16.8 | 9 | 2.4 |
| | college and above | 92 | 24.1 | 4 | 1.1 |
| Father's Occupation | Government employee | 148 | 38.8 | | |
| | Farmer | - | - | 360 | 95.7 |
| | Daily laborer | 31 | 8.1 | - | - |
| | Merchant | 207 | 53.0 | 16 | 4.3 |
| Mother's educational status | Unable to read and write | 78 | 20.5 | 218 | 58 |
| | Read and write | 103 | 27 | 130 | 34.6 |
| | Primary school | 79 | 20.7 | 28 | 7.4 |
| | secondary school | 64 | 16.8 | - | - |
| | college and above | 57 | 15 | - | - |
| Mother's occupation | Government employee | 69 | 18.1 | - | - |
| | Housewife | 216 | 56.7 | 323 | 85.9 |
| | Farmer | - | - | 53 | 14.1 |
| | Merchant | 96 | 25.2 | - | - |
| Family size | < or equal to 5 | 160 | 42.1 | 124 | 33.1 |
| | Greater than 5 | 220 | 57.9 | 251 | 66.9 |
| Wealth class | Lower class | 39 | 10.9 | 104 | 27.7 |
| | Middle class | 296 | 77.7 | 139 | 37.0 |
| | Upper class | 46 | 12.1 | 133 | 35.4 |

lower wealth class group were 2.2 times more likely to be stunted as compared to adolescent girls in the upper wealth class group [AOD: 2.24 (95% CI: 1.01, 4.86)] (**Table 5**).

## Factors associated with thinness

The regression analysis revealed that rural adolescent girls s who were in the middle adolescent age were 3 times more likely to be thin as compared to adolescent girls in the late adolescent age [AOR: 3.67 (95% CI: 1.20, 11.15)] (**Table 6**).

Urban adolescent girls who were in the early age were 8 times more likely to be thin as compared to late adolescent age [AOR: 8.38 (95% CI: 2.48, 28.30)] (**Table 7**)

The regression model of overall adolescent girls who were from farmer mothers were 2.8 times more likely to be thin as compared to adolescent girls from merchant mothers [AOR: 2.85 (95% CI: 1.05, 7.71)]. Adolescent girls who were in the early and middle adolescence age

**Table 2. Nutritional, health-related, and environmental characteristics among school adolescent girls in Debark district, 2020 (n = 757).**

| Variables | Response | Urban (381) rural (376) | | | |
|---|---|---|---|---|---|
| | | Frequency | % | Frequency | % |
| Food frequency | Less than 3 | 139 | 36.5 | 161 | 42.8 |
| | 3 and above | 242 | 63.5 | 215 | 57.2 |
| Dietary diversity | Adequate | 270 | 70.9 | 93 | 24.7 |
| | Inadequate | 111 | 29.1 | 283 | 75.3 |
| Food security | Food secure | 320 | 84 | 94 | 25 |
| | Food insecure | 61 | 16 | 282 | 75 |
| Menstruation began | Yes | 195 | 51.2 | 190 | 50.5 |
| | No | 186 | 48.8 | 186 | 49.5 |
| Age at first menstruation | 11 to 13 | 31 | 15.9 | 10 | 5.3 |
| | 14 to15 | 136 | 69.7 | 120 | 63.2 |
| | 16 to 17 | 28 | 14.4 | 60 | 31.6 |
| Diarrhea | Yes | 18 | 4.7 | 26 | 6.9 |
| | No | 363 | 95.3 | 350 | 93.1 |
| Mass media | Not exposed | 98 | 25.7 | 272 | 72.3 |
| | Lowest exposed | 50 | 13.1 | 22 | 5.9 |
| | Moderately exposed | 103 | 27 | 47 | 12.5 |
| | Very often exposed | 130 | 34.1 | 35 | 9.3 |
| Source of drinking water | Improved | 381 | 100 | 312 | 83 |
| | Unimproved | | | 64 | 17 |
| Distance of water source | < or equal to 30 | 381 | 100 | 291 | 77.4 |
| | Greater than 30 | | | 85 | 22.6 |
| Availability of home latrine | Yes | 337 | 88.5 | 172 | 45.7 |
| | No | 44 | 11.5 | 204 | 54.3 |
| Hand washing after toilet | Yes | 232 | 60.9 | 142 | 37.8 |
| | No | 149 | 39.1 | 234 | 62.2 |
| Availability of waste disposal | Yes | 290 | 76.1 | 126 | 33.5 |
| | No | 91 | 23.9 | 250 | 66.5 |

were 4.7 [AOR: 4.73 (95% CI: 2.05, 10.91)] and 2.9 [AOR: 2.95, 95% CI: 1.26, 6.87)] times more likely to be thin, respectively, as compared to adolescent girls in the late adolescence age. Adolescent girls who were in the middle wealth class group were 1.9 times more likely to be thin as compared to adolescent girls in the upper wealth class group [AOR: 1.97 (95% CI: 1.04, 3.72)] (**Table 8**)

## Discussion

This study illustrated the prevalence and associated factors of stunting and thinness among urban and rural adolescent girls. Accordingly, the overall prevalence of stunting and thinness was 20.1% (95% CI: 17.3, 23) and 10.3% (95% CI: 8.2%, 12.7%) respectively. Stunting was 24.2% and 16% among rural and urban adolescent girls, respectively. Besides, 12.1% of urban and 8.5% of rural girls were thin.

The overall prevalence of stunting was lower than studies conducted in West Bengal, India 32.5% [33], Rajabanshi, India 39% [34], Gondar 31.1% [35]. The West Bengal and Rajabanshi India's studies enrolled entirely participants aged from 10 to 14 years-old who are believed to be highly prone to stunting [36]. As to the Gondar's study, there is a time gap between the current study versus study conducted in Gondar. In the meantime, there was

**Table 3. Bivariable and multivariable logistic regression of factors associated with stunted among rural school adolescent girls, 2020 (n = 376).**

| Variable | Response | Stunted | | COR (95% CI) | AOR (95% CI) |
|---|---|---|---|---|---|
| | | No | Yes | | |
| Education status of the mother | Unable to read and write | 154 | 64 | **3.463(1.0101–11.878)** | 2.633(0.729–9.512) |
| | read and write | 106 | 24 | **1.887(0.526–6.761)** | 2.028(0.533–7.719) |
| | Primary school | 25 | 3 | 1.0 | 1.0 |
| Availability of latrine | No | 138 | 66 | **2.812(1.679–4.709)** | **1.954(1.114–3.427)** * |
| | Yes | 147 | 25 | 1.0 | 1.0 |
| you wash your hands after visited Toilet | No | 165 | 69 | 0.436(0.257–0.748) | 1.187(0.603–2.334) |
| | Yes | 120 | 22 | 1.0 | 1.0 |
| Availability of waste disposal | No | 177 | 73 | **2.475(1.404–4.370)** | 1.309(0.659–2.598) |
| | Yes | 108 | 18 | 1.0 | 1.0 |
| media exposure | not exposed | 198 | 74 | **3.987(1.185–13.412)** | 2.889(0.882–10.158) |
| | lowest exposed | 13 | 9 | **7.385(1.720–31.703)** | **5.135(1.159–22.741)** * |
| | moderately exposed | 42 | 5 | 1.270(0.282–5.711) | 1.216(0.265–5.586) |
| | very often exposed | 32 | 3 | 1.0 | 1.0 |
| Food security | Food secure | 77 | 17 | 1.0 | 1.0 |
| | Food insecure | 208 | 74 | 1.611(0.895–2.902) | 1.453(0.773–2.731) |
| Age category | Early | 77 | 29 | 1.841(0.949–3.573) | 1.423(0.655–3.093) |
| | Middle | 120 | 44 | 1.793(0.971–3.311) | 1.304(0.664–2.563) |
| | Late | 88 | 18 | 1.0 | 1.0 |
| Distance source water | ≤ 30 | 226 | 65 | 1.0 | 1.0 |
| | >30 | 59 | 26 | 1.532(0.895–2.623) | 0.902(0.494–1.647) |
| Adolescent education | Primary | 216 | 77 | 1.865(0.976–3.564) | 1.472(0.699–2.731) |
| | Secondary | 68 | 13 | 1.0 | 1.0 |
| Wealth index | Lowest class | 71 | 33 | **3.171(1.647–6.108)** | **2.583(1.310–5.091)** ** |
| | Middle class | 98 | 41 | **2.855(1.526–5.559)** | **2.366(1.230–4.554)** ** |
| | Upper class | 116 | 17 | 1.0 | 1.0 |

* = p value<0.05

** = p value < 0.01, p-value of Hosmer and Lemeshow test: 0.903.

implementation of safetynet program and Sequota Declaration (a program designed to fight against malnutrition and its determinants in Ethiopia). Those interventions could reduce the findings in this study.

**Table 4. Bivariable and multivariable logistic regression of factors associated with stunted among urban school adolescent girls, 2020 (n = 381).**

| Variable | Response | Stunted | | COR (95% CI) | AOR (95% CI) |
|---|---|---|---|---|---|
| | | No | Yes | | |
| Occupation of father | Government employee | 130 | 18 | 0.661(0.358–1.220) | 0.667(0.358–1.244) |
| | Daily laborer | 23 | 8 | 1.660(0.686–4.014) | 1.401(0.566–3.473) |
| | Merchant | 167 | 35 | 1.0 | 1.0 |
| Age category | Early | 106 | 36 | **3.170(1.446–6.946)** | **3.170(1.446–6.946)** ** |
| | Middle | 130 | 16 | 1.149(0.485–2.719) | 1.149(0.485–2.719) |
| | Late | 84 | 9 | 1.0 | 1.0 |
| Adolescent education | Primary | 178 | 45 | **2.244(1.217–4.136)** | 1.018(0.374–2.774) |
| | Secondary | 142 | 16 | 1.0 | 1.0 |

** = p value<0.01, p-value of Hosmer and Lemeshow test: 0.86.

**Table 5. Bivariable and multivariable logistic regressions of overall factors associated with over stunting among school adolescent girls in Debark district, 2020 (n = 757).**

| Variable | Response | Stunted | | COR (95% CI) | AOR (95% CI) |
|---|---|---|---|---|---|
| | | No | Yes | | |
| Residence | Urban | 320 | 61 | 1.0 | |
| | Rural | 285 | 91 | **1.675(1.167–2.404)** | 0.542(0.189–1.549) |
| Education status of the mother | Unable to read and write | 222 | 74 | **2.381(1.035–5.480)** | 1.901(0.208–17.377) |
| | read and write | 190 | 43 | 1.617(0.686–3.810) | 1.682(0.185–15.304) |
| | Primary school | 89 | 18 | 1.445(0.565–3.695) | 2.771(0.294–26.145) |
| | Secondary school | 54 | 10 | 1.323(0.468–3.741) | 4.395(0.437–44.231) |
| | College and above | 50 | 7 | 1.0 | 1.0 |
| Availability of latrine | No | 172 | 76 | **2.517(1.750–3.622)** | 1.531(0.648–3.613) |
| | Yes | 433 | 76 | 1.0 | 1.0 |
| wash your hands after visited Toilet | No | 293 | 90 | **1.546(1.078–2.217)** | 0.651(0.337–1.255) |
| | Yes | 312 | 62 | 1.0 | 1.0 |
| Availability of west disposal | No | 253 | 88 | **1.913(1.334–2.743)** | 0.862(0.361–2.061) |
| | Yes | 352 | 64 | 1.0 | 1.0 |
| Media exposure | not exposed | 280 | 90 | **2.204(1.316–3.692)** | 1.901(0.695–5.199) |
| | lowest exposed | 54 | 18 | **2.286(1.132–4.617)** | 3.225(0.996–10.441) |
| | moderately exposed | 127 | 23 | 1.242(0.656–2.350) | 2.169(0.764–6.160) |
| | very often exposed | 144 | 21 | 1.0 | 1.0 |
| Food security | Food secure | 347 | 67 | 1.0 | 1.0 |
| | Food insecure | 258 | 85 | **1.706(1.192–2.442)** | **1.951(1.008–3.779)***  |
| Age category | Early | 183 | 65 | **2.263(1.380–3.711)** | 0.000(0.000–0.000) |
| | Middle | 250 | 60 | 1.529(0.933–2.506) | 0.899(0.432–1.871) |
| | Late | 172 | 27 | 1.0 | 1.0 |
| First menstruation | 11 to 13 | 38 | 3 | 1.079(0.256–4.546) | 2.101(0.470–9.383) |
| | 14 to 15 | 213 | 43 | **2.759(1.131–6.727)** | **3.048(1.223–7.6)***  |
| | 16 to 17 | 82 | 6 | 1.0 | 1.0 |
| Distance source water | ≤30 | 546 | 126 | 1.0 | 1.0 |
| | >30 | 59 | 26 | **1.910(1.158–3.150)** | 1.660(0.649–4.247) |
| Adolescent education | Primary | 394 | 122 | **2.242(1.447–3.475)** | 1.523(0.808–2.871) |
| | Secondary | 210 | 29 | 1.0 | 1.0 |
| Water treatment | Untreated | 251 | 78 | **1.487(1.040–2.124)** | 0.855(0.325–2.252) |
| | Treated | 354 | 74 | 1.0 | 1.0 |
| Wealth index | Lowest class | 165 | 76 | **2.948(1.785–4.867)** | **2.215(1.008–4.864)***  |
| | Middle class | 280 | 51 | 1.166(0.696–1.954) | 0.581(0.228–1.483) |
| | Upper class | 160 | 25 | 1.0 | 1.0 |

* = p value<0.05, p value of Hosmer and Lemeshow test: 0.353.

Nevertheless, it is higher than the finding from South Asia, 11.1% [37]. Besides, it is also higher than a study done in Nigeria 10% [38]. In Nigeria's study most of the study participants were from urban residents that would impose less risk to develop stunting associated with better dietary practice, minimal work load, and awareness about feeding [9]. Furthermore, it's higher than the finding studies done in Eastern Ethiopia Balile 15% [39]. This might be due to a variation in cultural practices and socio-demographic characteristics.

Likewise, the study revealed that thinness was 10.3% (95% CI: 8.2%, 12.7%). This finding is consistent with a study conducted in Aksum, Ethiopia 12.6% [13]. However, it is lower than a finding from South Asia 38.6% [37], Hyderabad, Pakistan 27% [40], and Islamabad 46% [41].

**Table 6. Bivariable and multivariable logistic regression of factors associated with thinness among rural school adolescent girls, 2020 (n = 376).**

| Variable | Response | Thinness | | COR (95% CI) | AOR (95% CI) |
|---|---|---|---|---|---|
| | | No | Yes | | |
| DDS | Inadequate | 256 | 27 | 1.856(0.694–4.968) | 1.658(0.605–4.544) |
| | Adequate | 88 | 5 | 1.0 | 1.0 |
| Age category | Early adolescent | 97 | 9 | 2.366(0.705–7.935) | 2.753(0.812–9.330) |
| | Middle adolescent | 145 | 19 | **3.341(1.104–10.114)** | **3.665(1.205–11.149)**\* |
| | Late adolescent | 102 | 4 | 1.0 | 1.0 |
| Distance source water | ≤30 | 263 | 28 | 1.0 | 1.0 |
| | >30 | 81 | 4 | 0.464(0.158–1.362) | 0.403(0.136–1.194) |
| Wealth index | Lowest class | 93 | 11 | 1.092(0.468–2.548) | 1.118(0.458–2.730) |
| | Middle class | 131 | 8 | 0.564(0.226–1.407) | 0.586(0.231–1.484) |
| | Upper class | 120 | 13 | 1.0 | 1.0 |

\* = p value<0.05, p-value of Hosmer and Lemeshow test: 0.343.

In Pakistan's study adolescence in menstruation is prone to different risk factors like under-nutrition [42]. Besides, it is lower than finding from West Bengal, India 20.2% [33], and Rajabanshi, India16% [34]. The possible reason for the difference might be in the study of West Bengal and Rajabanshi, India were used early aged adolescents as a study participant, as a

**Table 7. Bivariable and multivariable logistic regression of factors associated with thinness among urban school adolescent girls, 2020 (n = 381).**

| Variable | Response | Thinness | | COR (95% CI) | AOR (95% CI) |
|---|---|---|---|---|---|
| | | No | Yes | | |
| Father education | Unable to read and write | 20 | 3 | 1.230(0.310–4.888) | 1.112(0.267–4.631) |
| | Read and write | 90 | 21 | 1.913(0.851–4.302) | 1.658(0.719–3.824) |
| | Primary school | 86 | 5 | 0.477(0.156–1.454) | 0.511(0.163–1.597) |
| | Secondary school | 57 | 7 | 1.007(0.362–2.802) | 1.321(0.456–3.825) |
| | College and above | 82 | 10 | 1.0 | 1.0 |
| Age category | Early | 111 | 31 | **8.378(2.480–28.301)** | **8.378(2.480–28.301)**\*\*\* |
| | Middle | 134 | 12 | 2.687(0.737–9.789) | 2.687(0.737–9.789) |
| | Late | 90 | 3 | 1.0 | 1.0 |
| Mother education | Unable to read and write | 69 | 9 | 1.728(0.505–5.919) | 2.421(0.525–11.151) |
| | Read and write | 85 | 18 | 2.806(0.901–8.742) | 3.256(0.803–13.209) |
| | Primary | 72 | 7 | 1.288(0.359–4.627) | 1.512(0.356–6.430) |
| | Secondary | 56 | 8 | 1.893(0.538–6.657) | 2.520(0.614–10.339) |
| | College and above | 53 | 4 | 1.0 | 1.0 |
| Availability of latrine | No | 36 | 8 | 1.749(0.757–4.039) | 1.317(0.502–3.455) |
| | Yes | 299 | 38 | 1.0 | 1.0 |
| Wash hands after visited toilet | No | 135 | 14 | 0.648(0.333–1.260) | 0.613(0.305–1.232) |
| | Yes | 200 | 32 | 1.0 | 1.0 |
| Adolescent education | Primary | 183 | 40 | **5.537(2.286–13.412)** | 2.592(0.723–9.293) |
| | Secondary | 152 | 6 | 1.0 | 1.0 |
| Wealth index | Lower class | 37 | 2 | 0.775(0.123–4.890) | 1.256(0.170–9.285) |
| | Middle class | 255 | 41 | 2.305(0.683–7.775) | 1.606(0.431–5.986) |
| | Upper class | 43 | 3 | 1.0 | 1.0 |

\*\*\* = p value<0.00, p-value of Hosmer and Lemeshow test: 0.901.

**Table 8. Bivariable and multivariable logistic regressions of overall factors associated with overall thinness among school adolescent girls in Debark district, 2020 (n = 757).**

| Variable | Response | Thinness | | COR (95% CI) | AOR (95% CI) |
|---|---|---|---|---|---|
| | | No | Yes | | |
| Residence | Urban | 335 | 46 | 1.0 | 1.0 |
| | Rural | 344 | 32 | 0.677(0.421–1.090) | 0.803(0.337–1.913) |
| Father education | Unable to read and write | 136 | 15 | 0.949(0.408–2.207) | 1.134(0.401–3.201) |
| | Read and write | 266 | 36 | 1.164(0.554–2.443) | 1.190(0.477–2.969) |
| | Primary school | 127 | 8 | 0.542(0.206–1.428) | 0.532(0.181–1.564) |
| | Secondary school | 64 | 9 | 1.209(0.464–3.149) | 1.259(0.433–3.658) |
| | College and above | 86 | 10 | 1.0 | 1.0 |
| Occupation of mother | Government employee | 63 | 6 | 0.608(0.219–1.688) | 0.670(0.237–1.889) |
| | Housewife | 490 | 49 | 0.638(0.332–1.228) | 1.046(0.521–2.099) |
| | Farmer | 43 | 10 | 1.485(0.602–3.663) | **2.848(1.053–7.704)*** |
| | Merchant | 83 | 13 | 1.0 | 1.0 |
| Age category | Early adolescent | 208 | 40 | **5.275(2.308–12.056)** | **4.727(2.047–10.913)*** |
| | Middle adolescent | 279 | 31 | **3.048(1.315–7.063)** | **2.947(1.264–6.873)*** |
| | Late adolescent | 192 | 7 | 1.0 | 1.0 |
| Family size | ≤5 | 249 | 35 | 1.0 | 1.0 |
| | >5 | 428 | 43 | 0.715(0.445–1.147) | 0.813(0.495–1.335) |
| Adolescent education | Primary | 449 | 67 | **3.093(1.603-5.968)** | 1.798(0.830–3.895) |
| | Secondary | 228 | 11 | 1.0 | 0.137 |
| Wealth index | Lower class | 226 | 15 | 0.701(0.337–1.458) | 0.744(0.351–1.578) |
| | Middle class | 284 | 47 | 1.748(0.961–3.180) | **1.967(1.041–3.718)*** |
| | Upper class | 164 | 16 | 1.0 | 1.0 |

* = p value<0.05

*** = p value<0.001, p value of Hosmer and Lemeshow test: 0.403.

result of the early growth spurt seen in adolescent girls with a sudden increase in nutritional requirement in early age group [6]. Similarly, it is lower than finding from Southwest Nigeria 16% [43]. The Nigeria's study was done only in rural adolescent girls but this study was done in both urban and rural areas. Furthermore, it is lower than finding from Ethiopian studies, Babile district 21.6% [24], Goba 20.9% [39], Adwa 21.4% [11], and Bale zone 13.6% [44]. This might be due to a difference in barriers in thinness such as cultural differences and other socio-demographic characteristics. Nevertheless, it is higher than a study conducted in Nigeria 5% [38]. This is possibly due to the difference in the study area. The study was conducted in one of the Nigeria state situated in the equatorial rain forest region which may have good nutrition access. Further, it could be associated with food culture practices among the two nations.

Adolescent girls lived in rural area are highly affected by stunting, 24.2% (95% CI: 20.2, 29) than those who are lived in urban area 16.0%(95% CI: 12.3%-19, 4%). This finding is supported by studies carried out in different areas of the world: a study conducted in South Asia's ranges from 30 to 60% in rural whereas 15 to 39% in urban [37], west Bengal India 35.7% in rural and 29.0% in urban [33], other study in India ranges from 22.7% to 31.5% in rural and 17.6% to 29.4% in urban [45], Mizan tepi 40.95% in rural and 17.8% in urban [46].

This disparity might be due to adolescent girls in urban has a good intake of a diversified diet than their counterpart rural residents [47]. Besides, adolescent girls from rural area were more likely to live in a poor socioeconomic condition that might affect their dietary intake

[48]. Similarly, poor health-seeking behavior in the rural residents might limit them from gaining access tonutrition services [47]. Intestinal parasitic infection are also common in rural inhabitant that would lead them to malabsorption, anemia, and other problem [49]. In general, this finding indicated that adolescent girls from rural settings are the most vulnerable for stunting that could further lead them to complications.

Similarly, 12.1% (95% CI: 8.9%, 15.5%) adolescent girls in urban area have experienced thinness. Whereas, 8.5% (95% CI: 5.6–11.4) of them from rural areas are thin. This finding is supported by a study in Rajasthan, India [45]. However, it contradicts with studies done in Adwa and Guyint [11, 25]. This disparity might be due to post-harvest weight gain since the data collection period of this study was in the post-harvest season while a study was done in Adwa and Goba was in pre-harvest season [50].

Overall adolescent girls from food insecure households near to two times more likely to be stunted which is supported by other local reports in Amhara national region [23], Dabat [12], Jimma [51]. It's the fact that one of the underlying causes of under-nutrition is food insecurity [52, 53]. This implies food insecure individuals have been affected by stunting and exposing them to further complications like poor school performance and poor reproductive outcomes. It's advised that accessing nutritious food for the destitute community would savage the lives of adolescent girls and their upcoming generations too.

In this study, the overall adolescent girls aged between 14 and 15 at first menstruation were three times more likely to be stunted as compared to those aged 16–17 at first menstruation. This may be due to early initiation of menstruation increased iron losses for a prolonged time which leads to reduce bone composition and bone mass [42]. Moreover, adolescent girls in menstruation experience decreased appetite [54]. The concerned body shall halt stunting during this critical period through diversified strategies such as iron supplementation and food fortification.

The overall adolescent girls who were in the lower wealth class group were two times more likely to be stunted as compared to those who are in the upper. Besides the finding also showed that rural adolescent girls who were in the lower wealth class and middle wealth class groups were 2.5 and 2 times respectively more likely to be stunted as compared to those in the upper wealth class group. This may be due to a reduction in the quantity of foods consumed and/or the replacement of higher-priced foods which are often less nutritious. Over a prolonged period, such changes may have adverse consequences for nutrition and a grave measure should be in place [55].

Likewise, rural adolescent girls who had lowest media exposure were five times more likely to be stunted. This finding consistent with a study done in rural community of Amhara national region [23]. Better media exposure would allow them to get better information about healthy eating and the consequences of poor dietary practices [56]. And it indicates that adolescent girls who did not have access to media exposure in rural settings had not get reliable and clear nutritional information.

In this study, rural adolescent girls who had no home latrine were two times higher odds of being stunted than those who had a home latrine. This finding was in agreement with another study in the rural community of Tigray [14]. It's the fact that proper sanitation can reduce stunting by preventing diarrheal and parasitic diseases [57]. Thus, it's indicated that it's important to address environmental sanitation, particularly in a rural setting.

The finding also showed that the urban adolescent girls who were in the early adolescent age were three times more likely to be stunted as compared to those in the late adolescent age. This finding is in agreement with other studies in Bangladesh [58], Assam [59], kaver district, Nepal [60]. Early adolescence is explained by a rapid physical maturation and they would be stunted related with the imbalance requirement and supply [36, 61].

Correspondingly, overall adolescent girls who were from farmer mothers are near to three times more likely to be thin as compared to those adolescent girls from merchant mothers. A study conducted here in Ethiopia has shown a similar finding [25]. This could be farmer mothers are loaded by laborious work [62]. As well, lack of knowledge about adolescent nutrition [63].

In this study, urban adolescent girls who were in the early adolescent age were eight times more likely to be thin as compared to adolescent girls in the late adolescent age. This finding was in agreement with studies in India [64], and another study in urban area of India [33]. Likewise, rural adolescent girls who are in the middle adolescent age are three and half times more likely to be thin as compared to adolescent girls in the late adolescent age. This finding consistent with a study in a rural area of India [65], rural Bangladesh [58]. Girls tend to develop moderate to high levels of disorder of eating behavior refers to many disturbing eating patterns, affects the nutritional status of adolescent girls [66].

The overall adolescent girls from the family with middle wealth class are near to two times more likely to be thin as compared to those adolescent girls from the family upper wealth class. This finding is supported by another study in Aksum, Ethiopia [13]. Middle household wealth status reflects the poor long-term economic status of the household which leads to poor nutritional status [55].

## Limitation of the study

The data collection period was in fasting season by Christian followers it might affect food frequency and dietary diversity factors.

## Conclusion

The prevalence of stunting was higher among rural adolescent girls than urban adolescent girls unlike thinness that was higher among urban than rural adolescent girls. Lowest media exposure, availability of home latrine, and lower and middle wealth class are important factors contributing to stunting among rural adolescent girls. Early adolescent age is an important factor contributing to stunting among urban adolescent girls. Correspondingly, being middle-aged and early age adolescent girls are an important contributing factor for thinness among rural and urban adolescent girls respectively. It's essential in targeting resources appropriately to raise the nutritional status the poor and the most vulnerable groups, including adolescent girls. Nutrition related media program for adolescents should be addressed through school minimedia, particularly in rural residences. Latrine coverage should be increased. In addition to ongoing efforts to fight child under nutrition, Ethiopia's Federal Ministry of Health (FMOH) should focus on policies and strategies that favor females. Finally, better to do a longitudinal study about the reason for poor growth throughout the period of adolescence.

## Supporting information

**S1 Questionnaire. English version of questionnaire.**
(DOCX)

**S2 Questionnaire. Amharic version of questionnaire.**
(DOCX)

**S1 Data.**
(SAV)

## Acknowledgments

The authors acknowledge the University of Gondar for securing ethical clearance for this study. We would like to extend our gratitude to the Debark Town Education Office for their support and individual households. The authors also acknowledge data collectors and supervisors.

## Author Contributions

**Conceptualization:** Tewodros Getaneh Alemu, Amare Demsie Ayele.

**Data curation:** Tewodros Getaneh Alemu.

**Formal analysis:** Tewodros Getaneh Alemu, Addis Bilal Muhye, Amare Demsie Ayele.

**Methodology:** Tewodros Getaneh Alemu.

**Software:** Tewodros Getaneh Alemu.

**Supervision:** Addis Bilal Muhye.

**Writing – original draft:** Tewodros Getaneh Alemu, Amare Demsie Ayele.

**Writing – review & editing:** Tewodros Getaneh Alemu, Addis Bilal Muhye, Amare Demsie Ayele.

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
