## [Decision Letter · Decision Letter 0]

7 Apr 2021

PONE-D-21-06490

Undernutrition and Associated Factors among Adolescent Girls Attending School in the rural and urban districts of Debark town, Northwest Ethiopia: A community-based comparative cross-sectional study.

PLOS ONE

Dear Dr. Alemu,

Thank you for submitting your manuscript to PLOS ONE. After careful consideration, we feel that it has merit but does not fully meet PLOS ONE’s publication criteria as it currently stands. Therefore, we invite you to submit a revised version of the manuscript that addresses the points raised during the review process.

We look forward to receiving your revised manuscript.

Kind regards,

Madhavi Bhargava, MD

Academic Editor

PLOS ONE

Journal Requirements:

1. Please ensure that your manuscript meets PLOS ONE's style requirements, including those for file naming. The PLOS ONE style templates can be found athttps://journals.plos.org/plosone/s/file?id=wjVg/PLOSOne_formatting_sample_main_body.pdf and https://journals.plos.org/plosone/s/file?id=ba62/PLOSOne_formatting_sample_title_authors_affiliations.pdf

3. Please state in your methods section whether you obtained consent from parents or guardians of the minors included in the study or whether the research ethics committee or IRB approved the lack of parent or guardian consent.

4. Please include additional information regarding the survey or questionnaire used in the study and ensure that you have provided sufficient details that others could replicate the analyses. For instance, if you developed a questionnaire as part of this study and it is not under a copyright more restrictive than CC-BY, please include a copy, in both the original language and English, as Supporting Information. Moreover, please include more details on how the questionnaire was pre-tested, and whether it was validated.

5. We suggest you thoroughly copyedit your manuscript for language usage, spelling, and grammar. If you do not know anyone who can help you do this, you may wish to consider employing a professional scientific editing service.  

Additional Editor Comments (if provided):

Reviewers' comments:

Reviewer's Responses to Questions

**Comments to the Author**

1. Is the manuscript technically sound, and do the data support the conclusions?

Reviewer #1: Yes

Reviewer #2: Yes

2. Has the statistical analysis been performed appropriately and rigorously? 

Reviewer #1: Yes

Reviewer #2: Yes

3. Have the authors made all data underlying the findings in their manuscript fully available?

Reviewer #1: Yes

Reviewer #2: Yes

4. Is the manuscript presented in an intelligible fashion and written in standard English?

Reviewer #1: No

Reviewer #2: Yes

5. Review Comments to the Author

Reviewer #1: Comments on manuscript entitled” Undernutrition and Associated Factors among Adolescent Girls Attending School in the rural and urban districts of Debark town, Northwest Ethiopia: A community-based comparative cross-sectional study”. PONE-D-21-06490

The authors assessed the association between the prevalence of undernutrition and different socio-demographic factors among school going adolescent girls of Debark district, Northwest Ethiopia.

I have the following few comments…

1. In the title, the authors mentioned as Adolescent Girls Attending School in the rural and urban districts of Debark town. It is not clear, whether Debark is town or district and whether Debark has rural and urban districts.

2. In abstract, total number of adolescent girls was mentioned as 792 in methodology and 757 in results. Need to clarify the same. Correct the error in spelling of stunting.

3. Page 10, line 3. Delete you in “low resistance to infection due you to low immunity”

4. Authors reported in methodology as—Lottery method was used to select 20% of the 51 schools. Thus, accordingly we need 10 schools; however, a total 13 schools were covered. Similarly, authors have not selected the schools proportionately. Authors need to explain whether they selected the number of schools purposively or there is any rationale behind it.

5. Authors selected 396 adolescent girls each from rural and urban areas. Here also authors have not followed the PPS method for selection of adolescent girls.

6. Discussion part is exhaustive. Need to condense the same

Reviewer #2: The topic is interesting and the findings are unique, it is a study that shows originality of the subject.

In the abstract: the data collection should not be written as indepth interview since the study design is not qualitative, it is a quantitative study. It is mentioned in the conclusion (in the abstract) that food insecurity associated with stunting and thinness, however this variable is not mentioned in the results (in the abstract).

In the introduction: The reason for conducting the study in adolescent nutrition is justified, however it should be better to add justification for investigating the disparities between urban and rural adolescent girls. Why we need to examine the disparities, what previous research tell about this? Is disparity between urban and rural become a big problem in Ethiopia?

In the method: The operational definition section provides detail explanation about each variable. The difference between moderately exposed and very often exposed is not clear, the cut off value for very often needs to be added.

Assessment of MDD requires respondents to report consumption of 10 food items, it needs explanation on what basis or consideration to choose/generate this list of foods?

The explanation of upper, middle, and lower class based on the quantiles in method section is not consistent with that described in narration of table 3 and 5 in the result section.

In the result: There is an error of typing drinking (dirking) water in table 2. Narration for table 3 mentions that adolescent girls who were in the lower and middle class experience stunting more than the upper class, referring the wealth index in the table (1st, 2nd, and 3rd quantile). This needs to be rechecked since in the method section, it is mentioned that 1st quantile as upper class, 2nd quantile as middle, and 3rd quantile as the lowest. It seems there is inconsistency for labelling the wealth class. The same problem with the narration of table 5. In table 6, wealth index uses different name of categories (i.e. lower, middle, late), I recommend to use same name of categories across tables.

In discussion: please provide practical implications based on the findings, for instance implication for policy or programs. I recommend to indicate contribution of this work to the scientific community.

6. PLOS authors have the option to publish the peer review history of their article (what does this mean?). If published, this will include your full peer review and any attached files.

Reviewer #1: No

Reviewer #2: No

---

## [Author Response · Author response to Decision Letter 0]

28 May 2021

Authors’ response to concerns of Editor and Reviewers

Dear Editor and Reviewers, we the authors of this article would be very happy to convey our deepest gratitude for your immense contribution - rigorous editorial and review concerns for which we will go one by one to make our manuscript suitable for publication in PLOS ONE journal. Therefore, we are going to respond the Editor, Reviewer #1 and Reviewer #2 concerns respectively as presented hereunder:

1. Editor concerns and Authors’ responses:

Editor concern: “1. Please ensure that your manuscript meets PLOS ONE's style requirements, including those for file naming.”

Authors’ response: Dear Editor, in all parts of the revised manuscript, we have used PLOS ONE’s requirements for publication of manuscripts. All the changes have been presented in a manuscript with track changes and without track changes. 

Editor concern: “2. Please provide additional details regarding participant consent. In the ethics statement in the Methods and online submission information, please ensure that you have specified what type you obtained (for instance, written or verbal, and if verbal, how it was documented and witnessed). If your study included minors, state whether you obtained consent from parents or guardians. If the need for consent was waived by the ethics committee, please include this information.

Once you have amended this/these statement(s) in the Methods section of the manuscript, please add the same text to the “Ethics Statement” field of the submission form (via “Edit Submission”).” 

Authors’ response: Dear Editor, in the revised manuscript, we have included the details of participant consent in the ‘Methods and Materials’ section. Previously, the ethical statement was written as a separate section; but now it is being moved to the methods part. Therefore, all the ethical statement queries and changes have been presented in a manuscript with track changes and without track changes.

Editor concern: “3. Please state in your methods section whether you obtained consent from parents or guardians of the minors included in the study or whether the research ethics committee or IRB approved the lack of parent or guardian consent.”

Authors’ response: Dear Editor, in the revised manuscript, we have included the details of consent from parents or guardians of the minors included in the study in the ‘Methods and Materials’ section. Previously, the ethical statement of consent from guardians or parents of minors was written as we took assent from them; but now it is being moved to the methods part and written explicitly as per your recommendation. Therefore, all the ethical statement queries and changes about assent from guardians or parents of minors have been presented in a manuscript with track changes and without track changes. 

Editor concern: “4. Please include additional information regarding the survey or questionnaire used in the study and ensure that you have provided sufficient details that others could replicate the analyses. For instance, if you developed a questionnaire as part of this study and it is not under a copyright more restrictive than CC-BY, please include a copy, in both the original language and English, as Supporting Information. Moreover, please include more details on how the questionnaire was pre-tested, and whether it was validated.” 

Authors’ response: Dear Editor, in the revised manuscript, we have included the information regarding the survey or questionnaire used in the study. We have stated the tool availability both in Amharic and English, its pretest and validation process before the actual data collection. Therefore, all the changes have been presented in a manuscript with track changes and without track changes.

Editor concern: “5. We suggest you thoroughly copyedit your manuscript for language usage, spelling, and grammar. If you do not know anyone who can help you do this, you may wish to consider employing a professional scientific editing service.” 

Authors’ response: Dear Editor, in the revised manuscript, we included an updated English language usage, spelling and grammar. We have given the manuscript for language edition and all the changes have been included in the manuscript with track changes and without track changes attached as supporting files. 

Editor concern: “Your ethics statement should only appear in the Methods section of your manuscript. If your ethics statement is written in any section besides the Methods, please move it to the Methods section and delete it from any other section. Please ensure that your ethics statement is included in your manuscript, as the ethics statement entered into the online submission form will not be published alongside your manuscript.” 

Authors’ response: Dear Editor, in the revised manuscript, the ethical statement it is being moved to the methods part only, and we have presented it in a manuscript with track changes and clean version. 

Editor concern: “Please review your reference list to ensure that it is complete and correct. If you have cited papers that have been retracted, please include the rationale for doing so in the manuscript text, or remove these references and replace them with relevant current references. Any changes to the reference list should be mentioned in the rebuttal letter that accompanies your revised manuscript. If you need to cite a retracted article, indicate the article’s retracted status in the References list and also include a citation and full reference for the retraction notice.”

Authors’ response: Dear Editor, in the revised manuscript, there is no any change that we made in the list of references. 

2. Reviewer #1 concerns and Authors’ responses:

Reviewer concern: “1. In the title, the authors mentioned as Adolescent Girls Attending School in the rural and urban districts of Debark town. It is not clear, whether Debark is town or district and whether Debark has rural and urban districts.”

Authors’ response: Dear Reviewer, when we say, in the rural and urban districts of Debark, we mean that Debark is a districts with rural and urban areas in its administration. 

Reviewer concern: “2. In abstract, total number of adolescent girls was mentioned as 792 in methodology and 757 in results. Need to clarify the same. Correct the error in spelling of stunting.”

Authors’ response: Dear Reviewer, in the abstract section of our manuscript, we mentioned 792 adolescent girls in the methodology part and while 757 in the results part. We would assume that these numbers are correctly stated in the abstract that 792 adolescent girls were the minimum sample size that was calculated while we design the research work. Whereas, on the field work, we actually collected the data from 757 adolescent girls that were included in the data analyses processes which would give us a response rate of 95.6%. That means 35 of our study participants were non-respondents.

The word ‘stunding’ in the original abstract is being corrected as ‘stunting’ in the revised version of the manuscript with track changes and without track changes. 

Reviewer concern: “3. Page 10, line 3. Delete you in “low resistance to infection due you to low immunity”

Authors’ response: Dear Reviewer, the word ‘you’ in the original manuscript is being deleted in the revised version of the manuscript with track changes and without track changes. 

Reviewer concern: “4. Authors reported in methodology as—Lottery method was used to select 20% of the 51 schools. Thus, accordingly we need 10 schools; however, a total 13 schools were covered. Similarly, authors have not selected the schools proportionately. Authors need to explain whether they selected the number of schools purposively or there is any rationale behind it.” 

Authors’ response: Dear Reviewer, as you clearly mentioned, if we took 20% of 51 schools, they would have been 10 schools to cover all the sampling issue. However, the number of schools from rural districts (we took 9 from 44 schools) gave us enough number of adolescent girls as per the proportional allocation; whereas taking proportionately calculated schools from the urban districts (we would have taken 1 from 7 schools), the number of adolescent girls there not enough to cover the required sample size from urban districts. Referring to the World Health Organization Guideline for Sampling Design for Community-based studies recommends that the proportion of study population should not be below 20%. It also recommends that there is a possibility to increase more than this proportion. Therefore, we took 4 schools in the urban districts to meet the minimum sample size required for our study. Finally, the 4 urban schools were selected by lottery method among the available 7 schools there. 

Reviewer concern: “5. Authors selected 396 adolescent girls each from rural and urban areas. Here also authors have not followed the PPS method for selection of adolescent girls.”

Authors’ response: Dear Reviewer, we have calculated the sample size of our study using a double population proportion formula where we used two different prevalence [Prevalence 1 (P1) for rural and Prevalence 2 (P2) for urban. This sample was calculated for a single group thereby we multiplied by two to get a sample of two settings. Moreover, we have used a 1:1 ratio of samples from rural and urban districts assuming that the we could get enough number of cases from the rural districts which was assumed have exposed for undernutrition. Finally, based on our ratio of samples, the overall sample size was divided equally for rural and urban districts. 

Reviewer concern: “6. Discussion part is exhaustive. Need to condense the same” 

Authors’ response: Dear Reviewer, the discussion part of our original manuscript is being condensed in the revised version of the manuscript with track changes and without track changes. 

3. Reviewer #2 concerns and Authors’ responses:

Reviewer concern: “In the abstract: the data collection should not be written as in-depth interview since the study design is not qualitative, it is a quantitative study. It is mentioned in the conclusion (in the abstract) that food insecurity associated with stunting and thinness, however this variable is not mentioned in the results (in the abstract).”

Authors’ response: Dear Reviewer, in the abstract section of our original manuscript, the phrase ‘in-depth interview’ is being replaced by ‘face-to-face interview’; and the variable ‘food insecurity’ is being included in the ‘results’ section of the abstract. Both are available in the revised version of the manuscript with track changes and without track changes 

Reviewer concern: “In the introduction: The reason for conducting the study in adolescent nutrition is justified, however it should be better to add justification for investigating the disparities between urban and rural adolescent girls. Why we need to examine the disparities, what previous research tell about this? Is disparity between urban and rural become a big problem in Ethiopia?” 

Authors’ response: Dear Reviewer, we have planned and conducted a comparative cross-sectional (urban and rural disparity) study assuming that there are higher exposure variables for undernutrition among rural adolescents. Previous evidences didn’t show us the clear variation in undernutrition among urban and rural settings. Moreover, nearly 80% of Ethiopian population have been living in the countryside areas in which the there is high fertility rates and low socioeconomic status. Therefore, in the revised version of the manuscript, we have included the points you raised in a manuscript with track changes and without track changes. 

Reviewer concern: “In the method: The operational definition section provides detail explanation about each variable. The difference between moderately exposed and very often exposed is not clear, the cut off value for very often needs to be added.”

Authors’ response: Dear Reviewer, the cut off value for ‘moderately exposed’ was ‘the highest value of all the three indicators is 2, i.e., at least once a week to any one or all of the indicators; whereas the cut of value for ‘very often exposed’ was ‘the highest value of all of the three indicators is 3, i.e., very often exposed to any one or all of the indicators. Therefore, in the revised version of the manuscript, we have included these cuts of values as in a manuscript with track changes and without track changes. 

Reviewer concern: “Assessment of MDD requires respondents to report consumption of 10 food items, it needs explanation on what basis or consideration to choose/generate this list of foods?” 

Authors’ response: Dear Reviewer, the Dietary Diversity Questionnaire (DDQ) were prepared based on the items presented in the Food and Agricultural Organization (FAO) Guideline 2016. Therefore, the list of food items was taken from this guideline and data were collected using this standard questionnaire. 

Reviewer concern: “The explanation of upper, middle, and lower class based on the quantiles in method section is not consistent with that described in narration of table 3 and 5 in the result section.” 

Authors’ response: Dear Reviewer, in the revised version of the manuscript, we have used a consistent categorization of wealth status as upper, middle and lower classes; and the changes were presented in a manuscript with track changes and without track changes. 

 Reviewer concern: “In the result: There is an error of typing drinking (dirking) water in table 2. Narration for table 3 mentions that adolescent girls who were in the lower- and middle-class experience stunting more than the upper class, referring the wealth index in the table (1st, 2nd, and 3rd quantile). This needs to be rechecked since in the method section, it is mentioned that 1st quantile as upper class, 2nd quantile as middle, and 3rd quantile as the lowest. It seems there is inconsistency for labelling the wealth class. The same problem with the narration of table 5. In table 6, wealth index uses different name of categories (i.e., lower, middle, late), I recommend to use same name of categories across tables.” 

Authors’ response: Dear Reviewer, in the revised version of the manuscript, we have changed the word ‘dirking’ as ‘drinking’ and in all section of the results part of the manuscript, we have used a consistent categorization of wealth status as upper, middle and lower classes; and the changes were presented in a manuscript with track changes and without track changes.

Reviewer concern: “In discussion: please provide practical implications based on the findings, for instance implication for policy or programs. I recommend to indicate contribution of this work to the scientific community.”

Authors’ response: Dear Reviewer, in the revised version of the manuscript, we have included the implication of our work for policy, program and scientific community. The changes were presented in a manuscript with track changes and without track changes.

---

## [Decision Letter · Decision Letter 1]

22 Jun 2021

Undernutrition and Associated Factors among Adolescent Girls Attending School in the rural and urban districts of Debark town, Northwest Ethiopia: A community-based comparative cross-sectional study.

PONE-D-21-06490R1

Dear Dr. Alemu,

We’re pleased to inform you that your manuscript has been judged scientifically suitable for publication and will be formally accepted for publication once it meets all outstanding technical requirements.

Kind regards,

Madhavi Bhargava

Academic Editor

PLOS ONE

Additional Editor Comments (optional):

Reviewers' comments:

Reviewer's Responses to Questions

**Comments to the Author**

1. If the authors have adequately addressed your comments raised in a previous round of review and you feel that this manuscript is now acceptable for publication, you may indicate that here to bypass the “Comments to the Author” section, enter your conflict of interest statement in the “Confidential to Editor” section, and submit your "Accept" recommendation.

Reviewer #2: All comments have been addressed

2. Is the manuscript technically sound, and do the data support the conclusions?

Reviewer #2: Yes

3. Has the statistical analysis been performed appropriately and rigorously? 

Reviewer #2: Yes

4. Have the authors made all data underlying the findings in their manuscript fully available?

Reviewer #2: Yes

5. Is the manuscript presented in an intelligible fashion and written in standard English?

Reviewer #2: Yes

6. Review Comments to the Author

Reviewer #2: All the reviewer's concerns have been addressed, and reviewer has no more additional comments to the authors

7. PLOS authors have the option to publish the peer review history of their article (what does this mean?). If published, this will include your full peer review and any attached files.

Reviewer #2: **Yes: **Rahayu Indriasari, PhD

---

## [Editor Report · Acceptance letter]

6 Aug 2021

PONE-D-21-06490R1 

Under nutrition and Associated Factors among Adolescent Girls Attending School in the  rural and urban districts of Debark, Northwest Ethiopia: A community-based comparative cross-sectional study. 

Dear Dr. Alemu:

I'm pleased to inform you that your manuscript has been deemed suitable for publication in PLOS ONE. Congratulations! Your manuscript is now with our production department. 

Kind regards, 

on behalf of

Dr. Madhavi Bhargava 

Academic Editor

PLOS ONE